# Short-term IL-15 priming leaves a long-lasting signalling imprint in mouse NK cells independently of a metabolic switch

Thuy T Luu[1], Laurent Schmied[1], Ngoc-Anh Nguyen[1], Clotilde Wiel[2], Stephan Meinke[1], Dara K Mohammad[1,3], Martin Bergö[2], Evren Alici[1,4], Nadir Kadri[1], Sridharan Ganesan[1], Petter Höglund[1]

**IL-15 priming of NK cells is a broadly accepted concept, but the dynamics and underlying molecular mechanisms remain poorly understood. We show that as little as 5 min of IL-15 treatment in vitro, followed by removal of excess cytokines, results in a long-lasting, but reversible, augmentation of NK cell responsiveness upon activating receptor cross-linking. In contrast to long-term stimulation, improved NK cell function after short-term IL-15 priming was not associated with enhanced metabolism but was based on the increased steady-state phosphorylation level of signalling molecules downstream of activating receptors. Inhibition of JAK3 eliminated this priming effect, suggesting a cross talk between the IL-15 receptor and ITAM-dependent activating receptors. Increased signalling molecule phosphorylation levels, calcium flux, and IFN-γ secretion lasted for up to 3 h after IL-15 stimulation before returning to baseline. We conclude that IL-15 rapidly and reversibly primes NK cell function by modulating activating receptor signalling. Our findings suggest a mechanism by which NK cell reactivity can potentially be maintained in vivo based on only brief encounters with IL-15 trans-presenting cells.**

## Introduction

NK cells are involved in the immune surveillance of malignant transformed and virally infected cells (1, 2, 3). Because of their expression of an array of activating and inhibitory germ-line encoded receptors, NK cells respond instantly against potentially dangerous cells and were originally classified as innate immune cells. More recently, NK cells have been shown to display adaptive features, including the requirement of accessory cells for an optimal functional response and the formation of memory-like cells after initial stimulation (4, 5, 6, 7).

IL-15 is a key cytokine for NK cell survival. Early findings showed that NK cells were absent from IL-15– or IL-15Rα–deficient mice, revealing the pivotal role of IL-15 in NK cell homeostasis (8, 9). More recent work has clarified that IL-15 controls several aspects of NK cell biology in vivo, such as proliferation, protection from apoptosis and effector functions (10, 11, 12). IL-15 acts mostly through *trans*-presentation, a process in which IL-15/IL-15Rα complexes on the surface of IL-15–producing antigen-presenting cells engage the IL-15Rβγ complex on responder NK cells (13). Binding of IL-15/IL-15Rα to the IL-15Rβγ complex induces different signalling pathways, including the JAK-STAT, PI3K-Akt-mTOR, and MAPK pathways (14, 15, 16).

In addition to controlling survival and homeostasis, IL-15 has also been implicated as a "priming factor" for NK cells, with the capacity to augment signalling pathways downstream of activating NK cell receptors (11, 17). This effect not only leads to better killing but also to increased migration to sites of inflammation and to better control of invading pathogens after TLR stimulation (11). Similarly, IL-15–primed NK cells showed increased cytotoxicity towards tumour cells and other targets (18, 19). Interestingly, NK cell priming seems to be reversible, as shown by the rapid loss of systemic functions of mature NK cells within a few days, following depletion of IL-15–presenting dendritic cells in vivo (10). These data suggest that IL-15 functions as a dynamic regulator of NK cell responsiveness in vivo. Clarifying the molecular mechanisms and kinetics of this regulation is important, not only to better understand NK cell function but also to improve the graft-versus-leukaemia effect after NK cell therapy of cancer patients. Identifying key environmental cues that regulate NK cell function in the new host is an important goal.

In this study, we addressed these questions by probing the cellular and molecular consequences of IL-15 priming in vitro, focusing on the early effects and their longevity. Our results show that IL-15 priming requires very little time in contact with the NK cell, resulting in enhanced NK cell function within 5 min after the start of the simulation. Importantly, short-term priming in our

[1]Department of Medicine Huddinge, Centre for Haematology and Regenerative Medicine (HERM), Karolinska Institutet, Huddinge, Sweden  [2]Department of Biosciences and Nutrition, Karolinska Institutet, Huddinge, Sweden  [3]Department of Food Technology, College of Agricultural Engineering Sciences, Salahaddin University-Erbil, KRG-Kurdistan Region, Iraq  [4]Cell Therapy Institute, Nova Southeastern University, Fort Lauderdale, FL, USA

Correspondence: petter.hoglund@ki.se

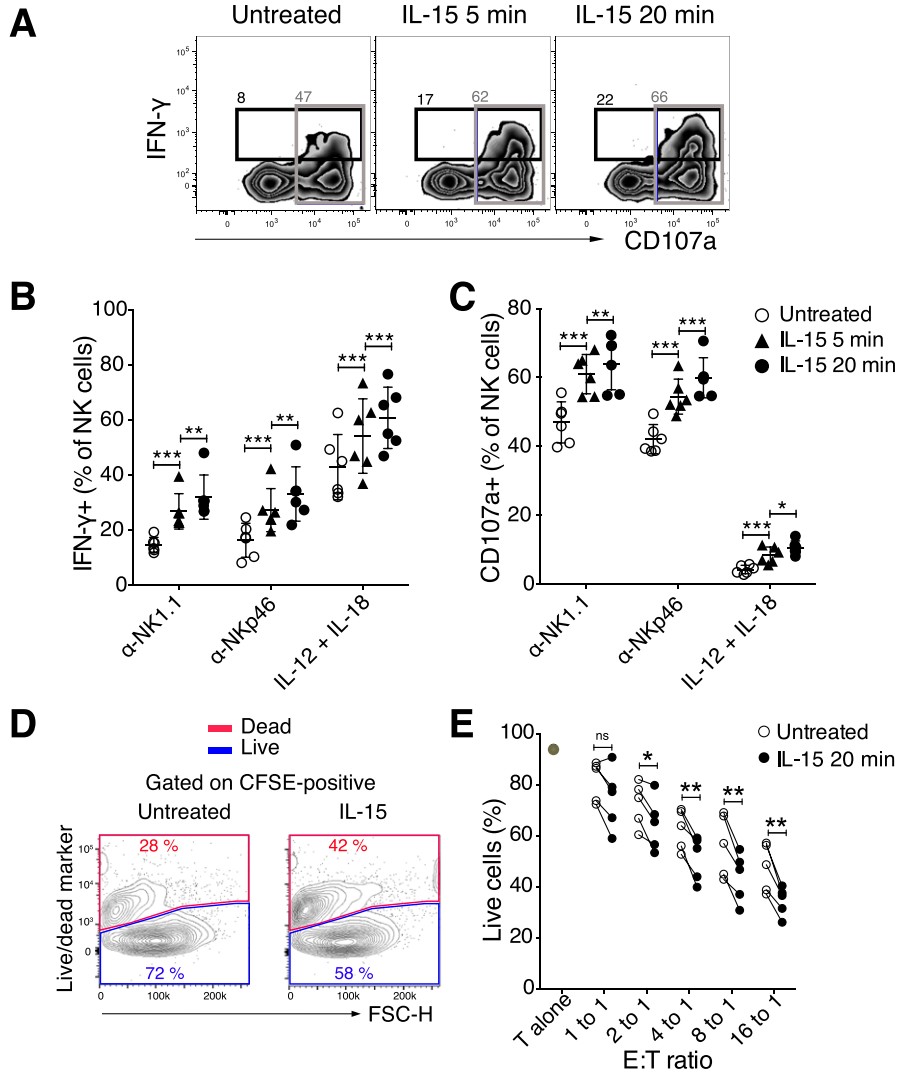

**Figure 1.  Short-term IL-15 stimulation primes IFN-γ production, degranulation, and cytotoxicity of NK cells.**
**(A, B, C, D, E)** Enriched NK cells were stimulated with 100 ng/ml IL-15 for 5 or 20 min, washed twice, and then assayed for IFN-γ production and degranulation after incubation on plate-bound α-NK1.1, α-NKp46, IL-12/IL-18, or killing capacity towards YAC-1 cells for 4.5 h (A) Representative plots and summary for (B) IFN-γ production and (C) degranulation in response to NK1.1 and NKp46 stimulation. Data were pooled from two independent experiments (n = 6). **(D)** Representative plots and (E) summary showing the killing capacity of NK cells towards YAC-1 target cells untreated or upon 20 min of IL-15 priming. YAC-1 cells were labelled with CFSE, and killing capacity was measured using a fixable live/dead marker after a 4 h 30 min of coculture. Data were pooled from two independent experiments (n = 5). T alone: target cells alone. **(B, C)** Pair-wise two-way ANOVA tests with Dunnett's multiple comparison test. **(E)** Paired *t* tests, error bars: SD, *P < 0.05, **P < 0.01, ***P < 0.001.

system was reversible, yet long-lasting, but did not depend on a metabolic imprint. Priming was mediated by a cross talk between IL-15 receptor signalling and immunoreceptor tyrosine-based activation motif (ITAM) signalling and involved rapid phosphorylation of signalling molecules in the ITAM-dependent signalling pathway. Our findings shed new light on the physiological phenomenon of NK cell priming and suggest new questions for a better understanding of NK cell function.

## Results

### Short-term IL-15 stimulation primes cytokine (IFN-γ) production, degranulation, and cytotoxicity of naive mouse NK cells

To study short-term IL-15 priming, NK cells were stimulated with IL-15 for 5 and 20 min in vitro, followed by measures of killing capacity and cytokine production. Variation in experimental conditions between mice was reduced by pooling NK cells from individual mice before cytokine

stimulation and downstream assays. Labelling with different doses of CellTrace Violet (CTV) as a "barcode" ensured analysis of individual mice in a pooled sample (Fig S1A). The barcoding itself did not affect IFN-γ production, degranulation or viability of the cells (Fig S1A and B). We found that as short as 5 min of IL-15 priming enhanced NK cell degranulation (CD107a) and/or cytokine production upon activating receptor crosslinking (NK1.1 and NKp46) or stimulation with IL-12+IL-18 (Figs 1A–C and S1C and D). IL-15 alone did not induce NK cell effector functions (Fig S1E), showing that the priming effect must be accounted for by the influence of IL-15 receptor engagement on activating signalling pathways.

Next, we investigated whether the increased degranulation that was observed in response to cross-linking of activating receptors translated into ameliorated cytotoxicity. Killing capacity and degranulation were evaluated in a combined assay, in which YAC-1 cells were cocultured with either IL-15–primed or naive NK cells (Fig 1D). 20 min of IL-15 treatment followed by extensive washing indeed augmented the killing capacity of NK cells towards YAC-1 target cells, as reflected by a significantly reduced fraction of live YAC-1

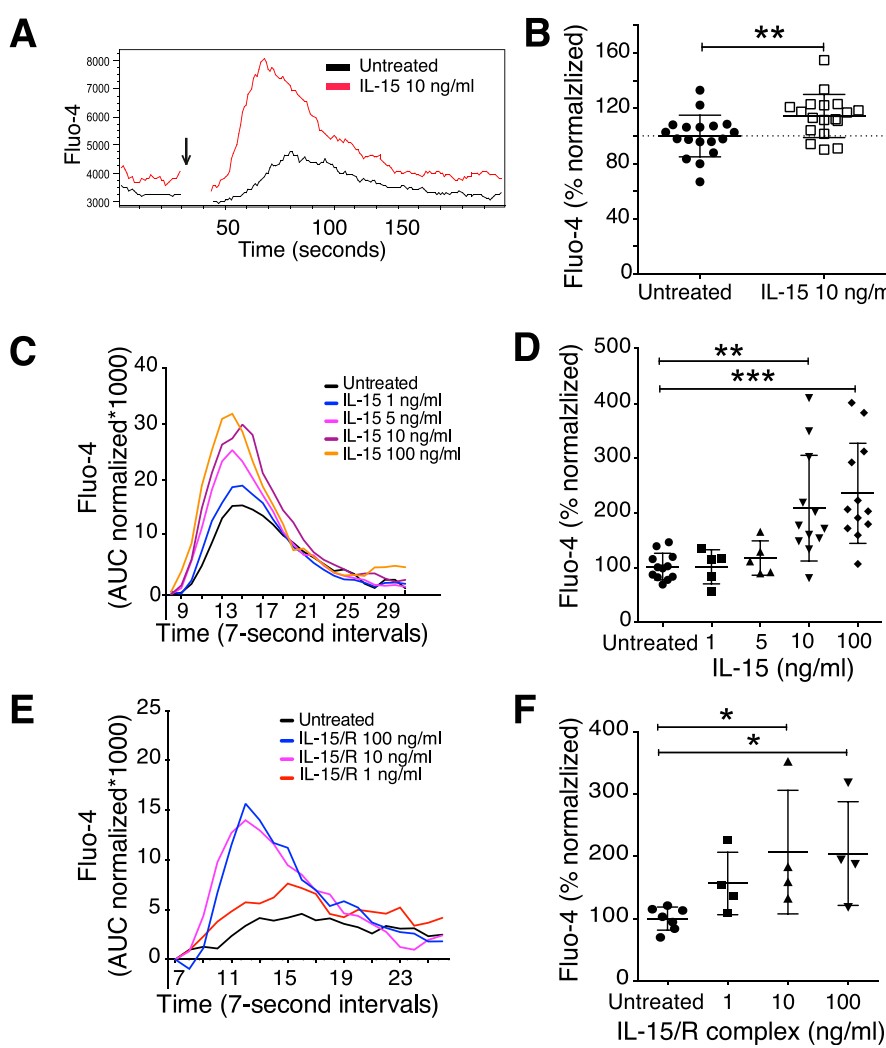

**Figure 2. Short-term IL-15 stimulation augmented calcium flux triggered via activating receptor NK1.1.**
**(A)** Representative plot of the calcium flux experiment. NK cells primed for 5 min were subjected to calcium flux assay after NK1.1 stimulation. Baseline calcium flux was measured for 30 s, then goat anti-mouse secondary antibody was added (indicated by the arrow), and finally, the samples were recorded for another 3 min. **(B)** Summary of baseline calcium flux. Y-axis: percentage of Fluo-4 area under the curve (AUC) before the secondary antibody was added in each sample over the average of corresponding values of untreated samples per experiment. Data were pooled from six independent experiments (n = 18). **(C, D)** Representative plot (C) and summary (D) showing the priming effect of different doses of soluble IL-15. Data were pooled from four independent experiments (n = 5–12). **(E, F)** Representative plot (E) and summary (F) showing the priming effect of different doses of IL-15/IL-15R complex. Data were pooled from two experiments (n = 4). **(C, E)** Y-axis: AUC of Fluo-4 signal normalized to the signal before secondary antibody was added. **(D, F)** Y-axis: Total AUC of Fluo-4 signal over the whole recording time normalized to the average of untreated samples in each experiment. **(B)** Paired $t$ test, (D, F) Kruskal–Wallis tests with Dunn's multiple comparison test, error bars: SD, *$P < 0.05$, **$P < 0.01$, ***$P < 0.001$.

cells after a 4, 5-h coculture with primed compared with naive NK cells (Fig 1D and E). The fraction of degranulating NK cells was correspondingly higher in IL-15–primed versus untreated NK cells, confirming the results in the activating receptor cross-linking experiments (Figs 1B and S1F).

The fast increase in NK cell function after IL-15 priming led us to explore whether proximal activating receptor signalling was affected. First, we investigated calcium flux responses in IL-15–primed and untreated NK cells (20). A similar barcoding as in the IFN-γ and degranulation assay was used, before IL-15 stimulation, to study calcium flux. The barcoding itself did not affect the efficiency of the dye loading process and the cell response towards activating receptor stimulation (Fig S2A). 5 min of IL-15 stimulation was sufficient to augment calcium flux in NK cells upon activating receptor crosslinking (Fig 2A, C, and D). An IL-15–dependent priming effect on calcium flux could be observed 1 min after IL-15 encounter (Fig S2B). A rapid priming response was also seen for human NK cells (Fig S2C and D), showing that short-term priming is not exclusive for murine NK cells.

Interestingly, IL-15 priming by itself increased the baseline $Ca^{2+}$ levels in NK cells (Fig 2A and B), suggesting that IL-15 may

have a direct impact on $Ca^{2+}$ channels, providing increased amounts of free intracellular $Ca^{2+}$ for second messenger signalling. To resemble a physiologically more relevant IL-15 presentation, we incubated NK cells for 5 min with IL-15/IL-15Rα complexes. The results showed comparably augmented calcium flux upon NK1.1 cross-linking (Fig 2E and F). IL-2 was also able to prime NK cells (Fig S2E and F), confirming the high degree of similarity between the two cytokines with regard to signalling pathways and cellular responses (21). Taken together, these data illustrate that short-term IL-15 priming improves proximal activation of NK cell signalling, which translates into enhanced functional responses including cytokine production, degranulation, and cytotoxicity.

## Phosphorylation of downstream signalling molecules of activating receptors

Based on the increase in the $Ca^{2+}$ flux response after short-term IL-15 treatment, we investigated phosphorylation of downstream activating receptor signalling molecules using a phosphoflow cytometry

protocol (22). To minimize variability between samples, cells from different stimulation groups were barcoded with a combination of two dyes and subsequently mixed before the analysis (Fig S3A). Untreated samples labelled with different barcoding levels showed similar phosphorylation of signalling molecules (Fig S3A and B), providing evidence that the barcoding process as such did not create undesired effects. 5 min of IL-15 stimulation increased phosphorylation of STAT5 and ERK1/2, which are well-known targets in the downstream signalling of the IL-15 receptor (Fig 3A and B). Moreover, elevated phosphorylation levels of molecules involved in activating receptor signalling, such as LCK and SLP-76, were seen (Fig 3C and D). The effect on activation of signalling molecules appeared broad with a wide range of molecules showing increased phosphorylation, including p38, STAT3, STAT4, and AKT (Fig S3B). Yet, the response was selective because phosphorylation levels of SRC, PLC-γ, ZAP70/SYK, and JNK were not increased (Figs 3C and D and S3B and C). Short-term IL-2 stimulation also induced phosphorylation of several of the same signalling molecules, but to a smaller extent than IL-15 (Fig S3D). Taken together, short-term IL-15 stimulation induces a selective activation of proximal signalling molecules downstream of ITAM-encoded activating receptors in NK cells.

Next, we asked whether priming with IL-15 would induce stronger phosphorylation of signalling molecules following activating receptor stimulation. To test this, NK cells were stimulated with IL-15 for 20 min following cross-linking of activating receptors and phosphoflow analysis. To avoid interference with the phosphoflow analysis, it was necessary to use a biotinylated primary antibody and streptavidin as a cross-linker. NKp46 stimulation induced phosphorylation of LCK and SLP-76, and these levels were indeed augmented by IL-15 stimulation (Fig 3E). Similar to our previous experiments, IL-15 alone did not induce phosphorylation of ZAP70/SYK and PLC-γ, yet a trend in augmentation was shown upon stimulation with both IL-15 and NKp46 also for these signalling molecules (Fig 3H). Surprisingly, the combination of the two stimulations also gave rise to a synergistic activation of signalling molecules downstream of IL-15, including STAT5 and ERK (Fig 3F), supporting a cross talk between these two pathways.

### Induction of mTOR signalling activation but not a metabolic shift

IL-15 stimulation is known to induce metabolic changes in NK cells, in particular through mTOR activation (18, 23). When we analysed mTOR and its downstream signalling molecules 4EBP1 and S6, we noted significantly increased phosphorylation of these molecules after short-term IL-15 treatment (Fig 3G and H). We next asked whether this enhanced mTOR signalling translated into metabolic changes. Using a Seahorse Analyzer, we assessed NK cell metabolism under three different conditions: In the first, NK cells were stimulated with 100 ng/ml IL-15, washed three times to remove unbound cytokine, and seeded on Seahorse plates for mitochondrial stress tests. This experiment revealed a decreased level of proton leakage and a trend in decreased respiratory capacity, as measured by maximal oxygen consumption rate (OCR) (Fig S4A and B). In the second

condition, unstimulated NK cells were seeded directly onto Seahorse plates, and after 20 min of baseline reading, IL-15 was added to a final concentration of 100 ng/ml. Consistent with the first setting, IL-15–treated samples exhibited lower oxidative phosphorylation (OXPHOS) along with a significantly decreased proton leakage (Fig 4A and B). Furthermore, we observed a trend of lower maximal respiratory capacity than control-treated samples (Fig 4A and B). Glycolysis remained unchanged for 20 min after IL-15 was added (Fig S4C).

In the third condition, we investigated the metabolism of NK cells treated with IL-15 for 22 h to validate our assay towards previous findings on NK cell metabolism after long-term IL-15 treatment. As opposed to short-term IL-15 stimulation, 22 h of IL-15 treatment increased both the OCR and extracellular acidification rate (ECAR) as compared with untreated freshly isolated NK cells (Fig 4C and D). The metabolic effects of long-term IL-15 stimulation were also reflected in an increase in the side scatter area (SSC-A), which was not seen for short-term stimulation (Fig S4D). A limitation in this third condition was the choice of control samples due to low survival of mouse NK cells in a 22-h culture without cytokines (Fig S4E). This problem was overcome by comparing metabolism data after 22 h of IL-15 stimulation with data from freshly isolated NK cells. Thus, in contrast to long-term priming, short-term priming had a limited metabolic impact, despite activation of the mTOR pathway.

### Steady-state reactive oxygen species (ROS) regulate phosphorylation of NK cell signalling molecules upon IL-15 treatment

OXPHOS is an important metabolic process for NK cell activation (24), and ROS, which are by-products of OXPHOS, are critical for Tcell activation by enhancing the activation of signalling molecules (25). ROS are well-known inducers of protein phosphorylation, primarily by means of phosphatase inhibition (26). To study whether ROS could be linked to augmented activation of signalling molecules in mouse NK cells (Fig 3), we investigated general phosphatase activity and phosphorylation of signalling molecules upon hydrogen peroxide ($H_2O_2$) treatment. As expected, $H_2O_2$ reduced tyrosine phosphatase activity in NK cell lysates in a colorimetric assay (Fig S4F). $H_2O_2$ also induced LCK and SLP-76 phosphorylation using phosphoflow analysis (Fig S4G). Of note, up to 200 μM of $H_2O_2$ for 5 min was not toxic for murine NK cells (Fig S4H).

To test whether augmented phosphorylation of signalling molecules after IL-15 treatment would be affected by scavenging ROS, we studied the activation of signalling molecules in NK cells at steady state and after 5 min of IL-15 treatment in the presence of N-acetyl cysteine (NAC). NAC is the precursor of glutathione and intracellular cysteine, and hence is used widely as an antioxidant (27). NAC treatment decreased IL-15–induced phosphorylation of STAT5, ERK1/2, LCK, and SLP-76, but not of SRC (Fig 4E and F). Notably, NAC alone showed a small but reproducibly reduced steady-state phosphorylation of ERK, SLP-76, and LCK, but not of STAT5 and SRC (Fig 4E and F). We also measured whether short-term IL-15 stimulation as such would induce ROS production. We found that ROS levels, measured indirectly by the

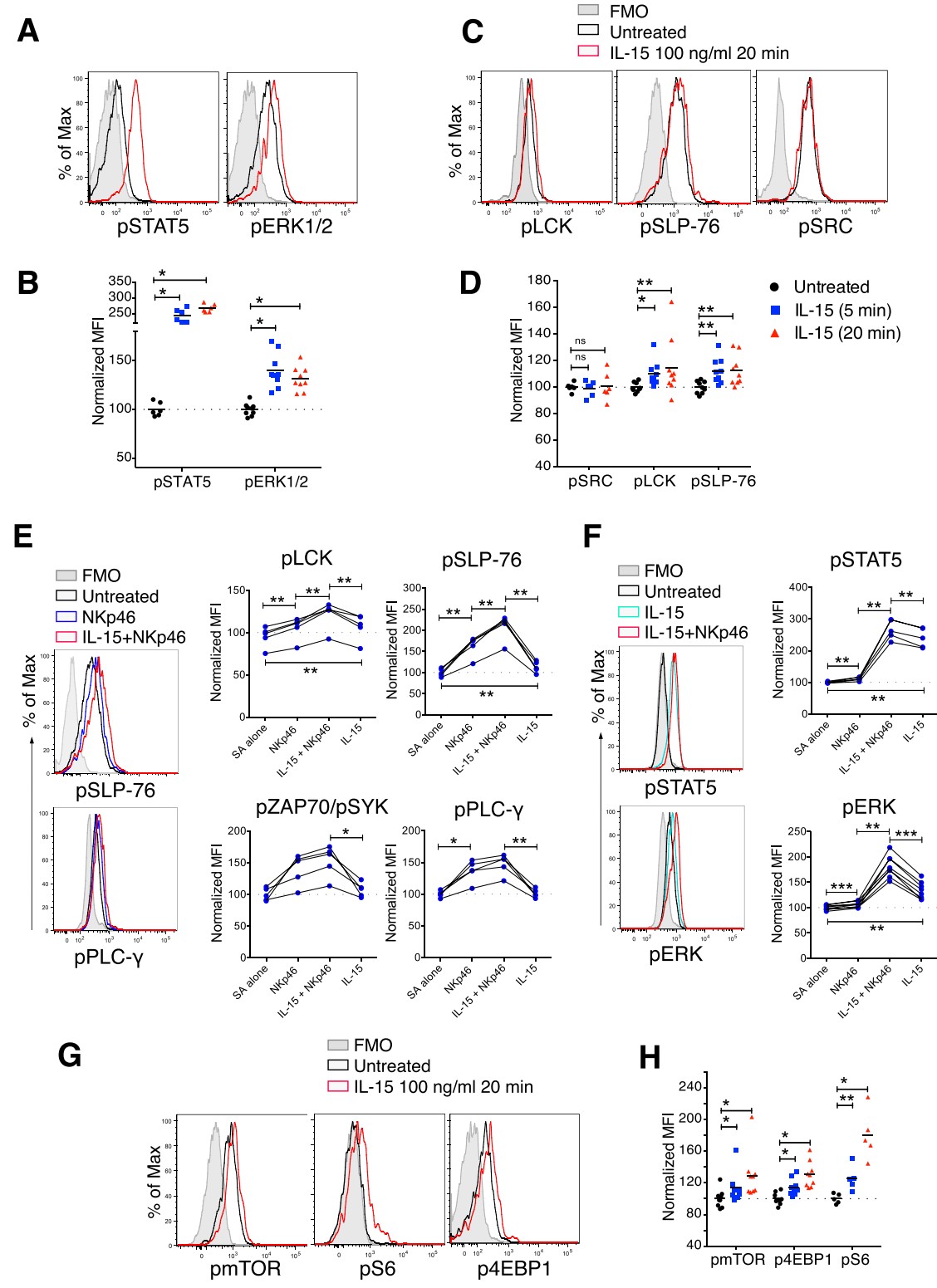

**Figure 3. Short-term incubation with IL-15 induces phosphorylation of activating and metabolism-related signalling molecules.**
Representative plots and summary graphs showing phosphorylation of signalling molecules upon short-term stimulation with 100 ng/ml IL-15. **(A, B)** Phosphorylation of IL-15–induced signalling molecules. **(C, D)** Phosphorylation of ITAM-linked signalling molecules. **(E, F)** Phosphorylation of (E) activating signalling molecules and (F) IL-15 signalling molecules after IL-15 stimulation alone, NKp46 stimulation alone, or IL-15+NKp46. Data were pooled from two to three independent experiments (n = 8). **(G, H)** mTOR-pathway signalling molecules. Data were pooled from two to four independent experiments (n = 5–11). **(B, D, E, F, H)** Pair-wise two-way ANOVA tests with Dunnett's multiple comparison test, ns, nonsignificant, *P < 0.05, **P < 0.01, ***P < 0.001.

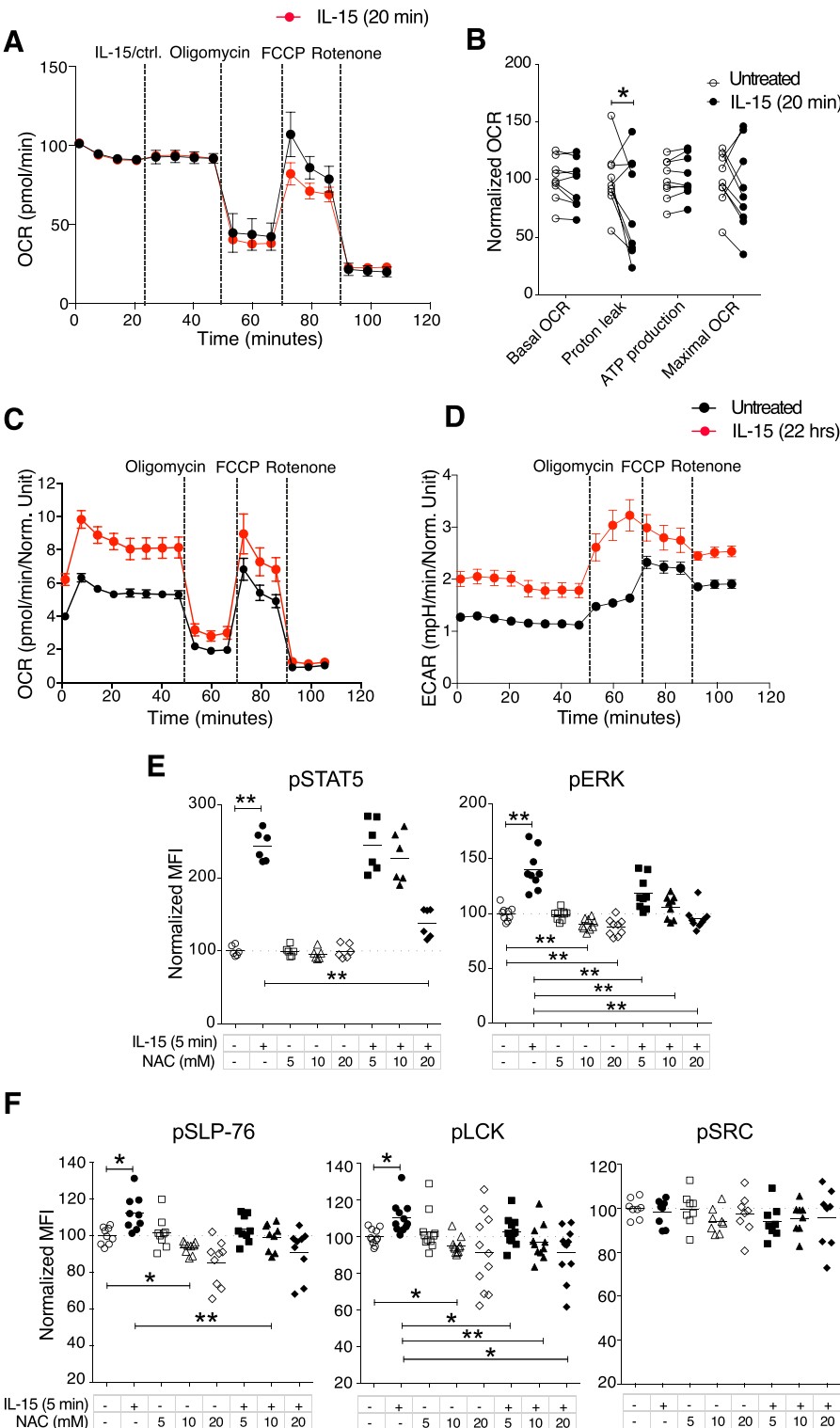

**Figure 4. Short-term IL-15 stimulation does not induce OXPHOS, yet reactive oxygen species level is important for activation of activating signalling molecules.**
**(A, B)** Representative data showing five replicates from one representative mouse and (B) summary of Seahorse experiments with acute IL-15 injections. Freshly isolated enriched NK cells were seeded onto a poly-L-lysine–coated Seahorse plate and IL-15 or a control solution was acutely injected at the indicated time point. Subsequently, baseline oxygen consumption rate (OCR) was measured for 20 min, and thereafter, oligomycin, FCCP, and rotenone were injected as indicated. Basal OCR, proton leak, ATP production, and maximal OCR values were normalized to the average of untreated samples in each experiment. Data were pooled from three independent experiments (n = 9). **(C, D)** OCR and (D) ECAR of 22-h IL-15 stimulated NK cells as compared with freshly isolated untreated cells from the same mice. Data show five replicates from one representative mouse from five mice from two independent experiments. **(E, F)** N-acetyl cysteine treatments and phosphorylation of STAT5, ERK1/2, SLP-76, LCK, and SRC at steady state and upon 5-min IL-15 stimulations. Data were pooled from three or four independent experiments (n = 8–11). **(B)** Pair-wise two-way ANOVA test with Dunnett's multiple comparison test. **(E, F)** Pair-wise one-way ANOVA tests with Holm–Sidak's multiple comparison test, ns, nonsignificant, *$P$ < 0.05, **$P$ < 0.01.

increase in fluorescence intensity of CM-H$_2$DCFDA (Fig S4I), indeed increased after 5 min of IL-15 stimulation, even if the magnitude of this increase was small and variable (Fig S4J and K). In summary, these data suggest that ROS may be part of the

modulation of phosphorylation of signalling molecules after short-term IL-15 priming. Further studies will be needed to pinpoint in more detail at which level of IL-15 signalling ROS scavenging operates.

## JAK3 inhibition reduces the IL-15 priming effect on NK cell effector functions triggered via activating receptor cross-linking

JAK3 is a key kinase in IL-15 signalling and has been implicated in the cross talk between IL-15 and NKG2D (28). To study whether JAK3 was involved in the priming effect upon short-term IL-15 stimulation, we made use of its inhibitor tofacitinib (CP-690,550), denoted here as iJAK3. We first confirmed that iJAK3 did not compromise NK cell viability (Fig S5A) and that it reduced the IL-15–mediated increase in phosphorylation of STAT5 and ERK, validating its expected biological effects (Fig 5A and B). We further found that iJAK3 reduced IL-15–induced phosphorylation of SLP-76 and LCK (Fig 5A and B), supporting our hypothesis that IL-15 signalling and ITAM signalling intersect. The inhibitory effect of iJAK3 on IL-15 priming extended to NK cell function because the effect of IL-15 priming on IFN-γ production and degranulation was abolished at 200 nM of iJAK3 (Figs 5C and S5B and C). Of note, iJAK3 treatment did not reduce responsiveness to activating receptor triggering in the absence of priming regarding IFN-γ production and degranulation (Figs 5C and S5B and C), providing suggestive evidence that NK1.1 signalling is qualitatively different in primed and unprimed NK cells.

## Long-lived signalling imprint in NK cells after short-term IL-15 stimulation

Physiological interactions between NK cells and DC in lymphoid tissues are short, being terminated within minutes (14, 15). This prompts the question how long primed NK cells would retain enhanced responsiveness after IL-15 priming. To address this question, we performed a 5-min IL-15 stimulation followed by extensive washes and subsequent incubation for 3 h in the absence of IL-15. 3 h after IL-15 removal, NK cells primed by 100 ng/ml, but not those primed with 10 ng/ml, still showed enhanced responsiveness as measured by Ca$^{2+}$ flux after cross-linking of NK1.1 (Fig 6A and B). Priming of IFN-γ production in response to NK1.1 stimulation was also detected after 3 h of IL-15 starvation (Fig 6C and D). Supporting these data, IL-15–induced pSTAT5 remained detectable for up to 1 h after washing but returned to pre-priming levels after 3 h (Fig 6E and F). These results suggest that the functional enhancement by short-term IL-15 priming is both long-lasting and reversible, while coinciding with a corresponding signalling imprint.

# Discussion

Our study has led to several novel conclusions of importance as to how IL-15 primes NK cell function and controls NK cell responsiveness. Firstly, we found that IL-15 primes NK cell function very quickly, leading to augmented effector responses after as little as 5 min of pre-incubation. Secondly, the short-term priming effect was mediated by JAK3 and ROS and was associated with increased steady-state phosphorylation levels of several activating receptor–associated signalling molecules. Thirdly, mTOR activation was increased, whereas the OXPHOS response remained unchanged, or was even reduced in some aspects, after up to 20 min of IL-15 stimulation. Finally, the priming effect remains for several hours after removal of

exogenous IL-15, reflected by the persistence of an augmented calcium flux response and enhanced levels of pSTAT5.

These findings can be interpreted in light of recent advances in understanding the cross talk between NK cells and accessory cells in lymph nodes, which are characterized by a series of "touch-and-go" interactions shorter than 5 min (14, 29). Thus, for NK cell priming to be relevant as an in vivo regulatory mechanism, a fast-acting mechanism such as the one described in this study is necessary. In line with our study, Marçais et al reported an increase in calcium flux of murine NK cells upon 30-min IL-15 stimulation (30). An important next question is how such a rapid touch of an NK cell–priming accessory cell effect would be regulated. A key finding in our study was a rapid activation of proximal ITAM-associated signalling molecules including LCK and SLP-76, which only partially overlap with the IL-15 signalling pathway (15, 16, 31). The effect on steady-state phosphorylation was broad, yet selective. Although we consistently identified activation of LCK and SLP-76, other signalling proteins such as PLC-γ, ZAP70/SYK, SRC, and JNK were not activated. In addition, we demonstrated the relevance of these small changes as IL-15 stimulation augmented the increased activation of LCK and SLP-76, which was induced by activating receptor stimulation.

The dissociation between the increase in the mTOR pathway activation, the decrease in the maximal respiratory rate, and the involvement of ROS in short-term IL-15 treatments was an unexpected combination of results. When it comes to the involvement of ROS, our data suggest that the induction of phosphorylation by IL-15 is ROS-dependent. However, because ROS induction by IL-15 was weak and variable and an effect of NAC was also seen in unprimed NK cells, we believe that our data point to a role of steady-state ROS, rather than IL-15–induced ROS in the regulation of signalling molecule phosphorylation. In this respect, it is interesting that the weak effect of NAC alone was only seen for molecules directly implicated in ITAM signalling and that STAT-5, which is uniquely targeted by IL-15 signalling, was not affected. This might imply a steady-state continuous balance between kinases and phosphatases in activating receptor signalling in NK cells, which is not seen in cytokine receptor signalling. We do not exclude, however, that ROS production induced by IL-15 receptor signalling is also involved. ROS might then be induced by an OXPHOS-independent mechanism involving NADPH oxidase (NOX) and superoxide dismutase (SOD) (32), similar to TCR-triggering in T cells (33).

The mTOR pathway, mainly through mTORC1, induces the glucose metabolism and OXPHOS through expression induction of metabolic enzymes and mitochondrial structure proteins (34). We found that mTORC1 and its downstream molecules 4EBP1 and S6 were increased after 5 and 20 min of IL-15 stimulation yet were not accompanied by an increase in the OXPHOS or glycolysis rate, as measured using Seahorse technology. One possible explanation for this discrepancy is that mTORC1-mediated metabolic changes may take longer than 20 min after IL-15 exposure, perhaps involving several coordinated processes including gene transcription, mRNA translation, and protein synthesis. We are also not sure at this point what the reduction in maximal OCR and proton leak after short-term IL-15 priming means in terms of the functional consequences that we observe. Future work will be needed to dissect the kinetic link between IL-15 priming and the induction of metabolic changes, which goes beyond the scope of the present study.

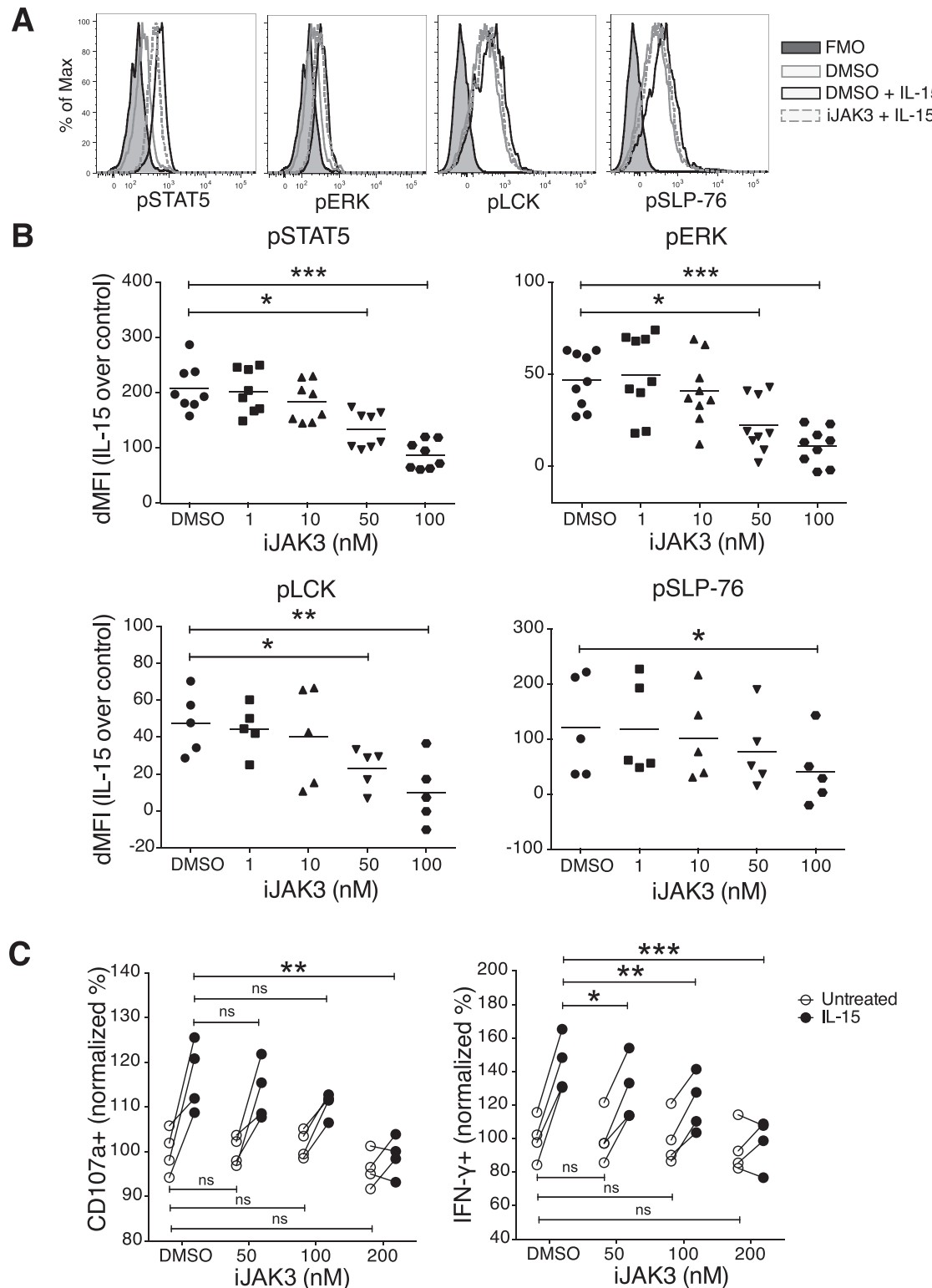

**Figure 5. Short-term IL-15 priming is mediated through JAK3.**
**(A, B)** Representative histograms for 200 nM iJAK and (B) summary of STAT5, ERK, and p38 phosphorylation upon iJAK3 treatments. Enriched NK cells were incubated with iJAK3 for 1 h before IL-15 was added directly to the culture to a final concentration of 100 ng/ml. After 5-min incubation, the cells were subjected directly to the phosphoflow protocol. The y-axis shows dMFI (MFI of IL-15–primed samples subtracted to the MFI of the corresponding unprimed samples treated with the same doses of iJAK3). Data were pooled from two independent experiments (n = 4–5). **(C)** IL-15 priming effect and iJAK3 inhibition on IFN-γ production (left panel) and CD107a expression (right panel). Enriched NK cells were primed with 100 ng/ml IL-15 for 5 and 20 min and washed three times to remove unbound cytokine. The primed and

Our finding that JAK3 inhibition prevented IL-15–induced phosphorylation of ITAM-associated signalling molecules suggests that the IL-15 receptor may be directly interacting with the activating receptor NK1.1 near the plasma membrane. Such an interaction would bring key components of the two receptors close to each other, providing grounds for such a cross talk, which is supported by the interaction between JAK3 and DAP10, an adaptor downstream of NKG2D (28). Furthermore, a pull-down study in combination with a proteomics analysis identified PLC-γ and ZAP70 as binding partners of IL-2Rγ/IL-15Rγ (35). IL-15 stimulation did not increase phosphorylation of ZAP70/SYK and PLC-γ as such, yet IL-15 resulted in an increased activation of these molecules after NKp46 stimulation. This supports our model suggesting a proximal cross talk between the IL-15 receptor and activating receptors. This process might also involve steady-state ROS, which in turn leads to an increased basal activation state and a higher likelihood for signalling initiation and induction of a heightened state of responsiveness. The circuit could also include IL-15–mediated induction of ROS, which further augments the enhanced activation of signalling molecules via inhibiting their breakdown by phosphatases. A role for ROS in the increased basal activation state was further supported in our study by the increased resting calcium $Ca^{2+}$ flux upon hydrogen peroxide treatment (data not shown), although we could not exclude that ROS acts directly on endoplasmic reticulum calcium channels, as also suggested (36). The role of IL-15–induced ROS, however, remains uncertain because it was not accompanied by a correspondingly enhanced metabolic state in our study.

A final new insight from our study relates to the reversible effect of NK cell priming, reflected by the finding that 5 min of IL-15 stimulation induced an enhanced functional response that lasted up to 3 h before waning. Similar notions were made before, but after much longer periods of IL-15 stimulation and starvation (18). Physiologically, our data may suggest that NK cells, after receiving a priming signal by IL-15, delivered by, for example, DCs (10), remember this encounter, during which time efficient effector functions can be performed at distant sites. In addition, because the priming effect was dose-dependent, the degree of enhanced responsiveness of the NK cells will depend on the strength of the priming signals received within the preceding hours. In this respect, it is interesting to note that a subset of ex vivo NK cells, the so-called serial killers, are most strongly active during the first 2–3 h, after which the killing activity decreases (37). One explanation for this decline in cytotoxicity could be the loss of priming. Whether or not a primed NK cell that has lost its priming memory can be primed again, thus repeating the cycle, remains to be investigated. A future question is also how NK cell priming by IL-15 cooperates with NK cell education based on different levels of MHC class I, as we suggested previously in a rheostat model for NK cell education (38). An interesting possibility is that these two levels of input cooperate in circulating NK cells, for them to display an activation threshold most suitable for the surrounding environment. In summary, our findings of short-term and reversible IL-15 priming shed new light on the in vivo functional priming of NK cells and provide a novel understanding of how NK cell effector functions are regulated.

# Materials and Methods

### Mice

All animal procedures were approved by the Animal Ethics Committee in Stockholm, Sweden (Stockholms Södra Djurförsöksetiska Nämnd and Linköpings Djurförsöksetiska Nämnd). The mice were housed at the animal facility at the Karolinska Institute, Huddinge, Sweden. Experimental mice were 6–10 wk old and were sex- and age-matched in each experiment. The mice used in this study are on an inbred C57Bl/6 background, deficient for the classical MHC class I genes H2K$^b$ and H2D$^b$ but transgenic for the H2D$^d$ gene. These mice were previously generated in our laboratory to provide a well-characterized genetic background for studies of NK cell education (38, 39, 40). All NK cells in these mice are functionally educated on one single MHC class I allele (38). Because of the its well-characterized NK cell system, these "single H2D$^d$ mice" provide a well-characterized model of NK cell function and priming.

### Antibodies and flow cytometry analysis

All surface antigen staining procedures were performed for 20 min at 4°C in FACS buffer (PBS + 2% FBS). Data were acquired from an LSR Fortessa flow cytometer (BD Biosciences) and analysed using FlowJo v9.9.6 (Tree Star). The antibodies used for surface staining include CD3 (145-2C11), NKp46 (29A1.4) from BD Biosciences, and NK1.1 (PK136) from BioLegend. Antibodies used for in vitro stimulation and killing assays include NKp46 (polyclonal; R&D Systems), NK1.1 (PK136; BioLegend), CD107a (1D4B; BD Biosciences), and IFN-γ (XMG1.2; BD Biosciences). Dead cells were determined using a LIVE/DEAD Fixable Aqua Dead Cell Stain Kit (Invitrogen). Rat anti-mouse antibodies used for mouse NK cell enrichment obtained from BioLegend include CD3 (17A2), CD19 (6D5), Ter119 (TER-119), Gr-1 (RB6.8C5), CD4 (RM4–5), and CD8 (53–6.7). The secondary antibody used to cross-link mouse NK1.1 antibodies in the calcium flux experiments is goat anti-mouse IgG (H + L) polyclonal antibody (Jackson Immunoresearch).

### Mouse NK cell isolation and barcoding for in vitro stimulation and calcium flux assays

NK cells were enriched from total splenic single-cell suspension by using a customized negative selection protocol. Briefly, splenic cells were stained with purified rat anti-mouse mAbs against CD3, CD19, Ter119, CD4, CD8, and Gr-1 for 15 min at 4°C. After washing once with

---

unprimed cells were placed in culture with plate-bound anti-NK1.1 antibody in the presence of indicated iJAK3 concentrations. The y-axis shows fractions of cells that produced IFN-γ or expressed CD107a normalized to the average of unprimed DMSO-treated samples in each experiment. Data were pooled from two independent experiments (n = 4). **(B)** Pair-wise Friedman test with Dunnett's multiple comparison test. **(C)** Pair-wise two-way ANOVA test with Sidak's multiple comparison test, ns: nonsignificant, *P < 0.05, **P < 0.01, ***P < 0.001.

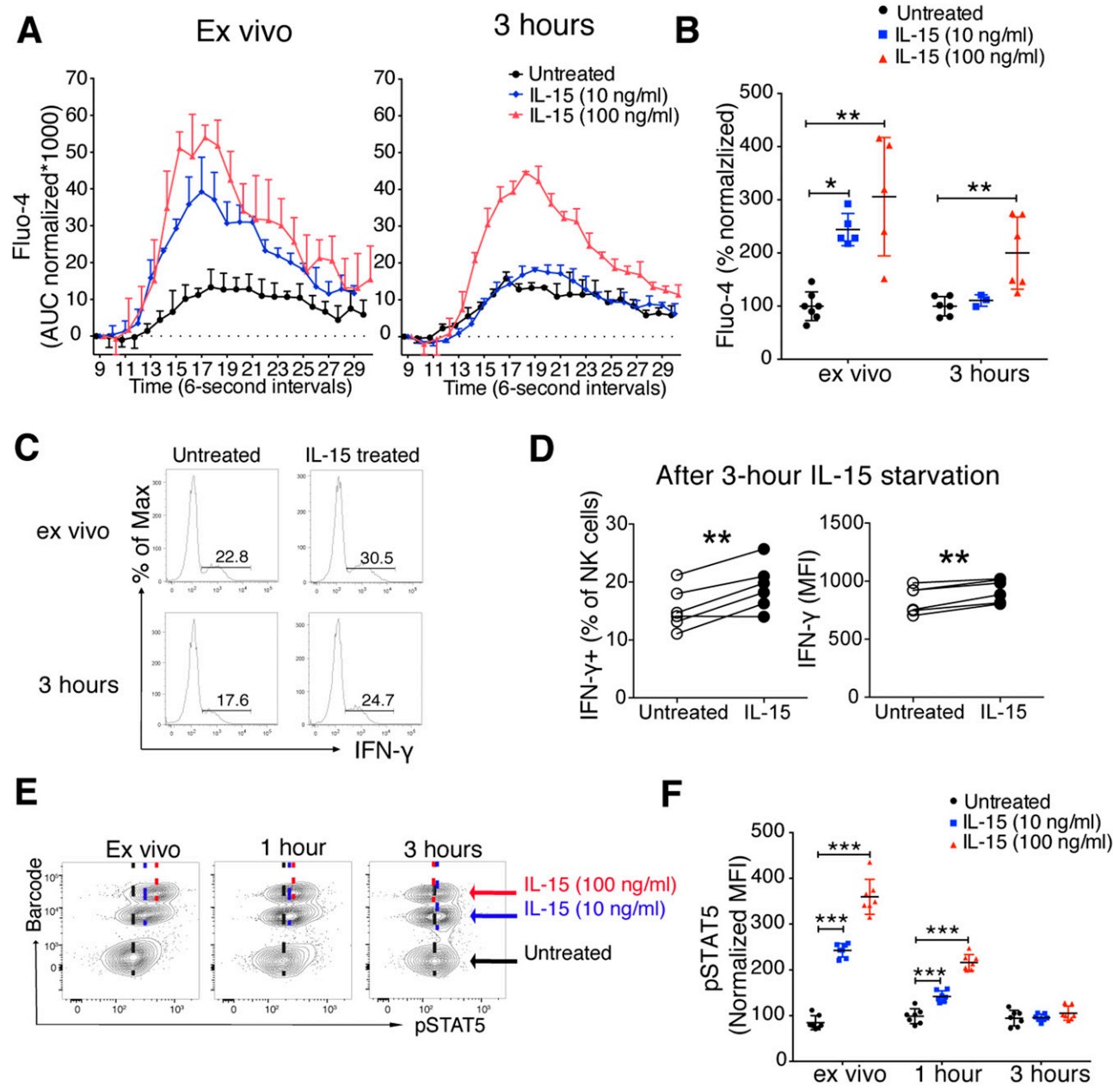

**Figure 6. Short-term IL-15 stimulation leaves long-lived signalling imprints in NK cells despite extensive washes.**
**(A, B)** Representative plot (A) and summary (B) showing the long-lasting effect of IL-15 priming on calcium flux response. Primed NK cells (5 min) were incubated for 3 h in the absence of IL-15 and subjected to calcium flux following NK1.1 stimulation. Data were pooled from three independent experiments (n = 3–7). **(A)** Y-axis: AUC of Fluo-4 signal normalized to the signal before secondary antibody was added. **(B)** Y-axis: Total AUC of Fluo-4 signal over the whole recording time normalized to the average of untreated samples in each experiment. **(C, D)** Representative plot (C) and summary (D) showing the long-lasting effect of IL-15 priming (100 ng/ml, 20 min) on IFN-γ production. Primed NK cells were incubated for 3 h in the absence of IL-15 and subjected to 4 h 30 min of NK1.1 stimulation with plate-bound anti-NK1.1 antibody. Data were pooled from two independent experiments (n = 6). **(E, F)** Representative plot (E) and summary (F) showing retained STAT5 phosphorylation after 20 min of IL-15 stimulation and 1-h IL-15 starvation. Data were pooled from two independent experiments (n = 7). **(B)** Kruskal–Wallis tests with Dunn's multiple comparison test, and comparisons were made between treatments in each time point. **(D)** Paired *t* test. **(F)** Pair-wise two-way ANOVA test with Sidak's multiple comparison test. *P < 0.05, **P < 0.01, ***P < 0.001.

ice-cold PBS + 2% FBS, cells were incubated with Dynabeads Sheep Anti-Rat IgG (Invitrogen) at 0.3 beads/cell for 15 min in a rotator at 4°C. Bead-bound cells were separated using a magnet. The unbound cells were transferred to a new tube that contained washed beads, and the process was repeated twice. Enriched NK

cells from individual mice were labelled with different doses of CellTrace Violet (Life Technologies) as a cell barcoding method. 4 × 10⁶ cells per ml were incubated with CellTrace Violet of different doses (generally 1.2, 0.3, and 0 μM) for 15 min at RT in PBS. Excess dye was washed away by three washes using RPMI + 10% FBS. Enriched

NK cells from different mice labelled with different concentrations of CTV were pooled and subjected to downstream assays in the same tubes.

## IL-15 priming and calcium flux experiment

Barcoded and enriched NK cells were incubated in vitro with purified mAbs against NK1.1. Cells were stimulated with rmIL-15 (Immuno-tools), rmIL-15/R complex (eBioscience), or rmIL-2 (Immunotools) in RPMI + 10% FBS. In some experiments, the cells were incubated at 37°C for up to 3 h in the absence of IL-15 before being subjected to the following steps. NK cells were subjected to surface staining and calcium assays, as previously described (41). The summary data of several experiments show the total AUC over the whole recording time normalized to the average of the total AUC of untreated samples in each experiment.

## IL-15 priming and in vitro stimulation assay

Enriched NK cells were stimulated with plate-bound α-NK1.1 or α-NKp46 antibodies, or a combination of 1 ng/ml IL-12 (p70; BD Biosciences) plus 1 ng/ml IL-18 (ImmunoTools). Stimulations and staining were performed, as described previously (10).

## JAK3 inhibition experiments

Enriched NK cells were primed and washed, as indicated earlier. iJAK3 (tofacitinib citrate, CP-690550; Sigma-Aldrich) was added directly into the 5-h culture of primed or naive enriched NK cells on the α-NK1.1–coated plate. The stimulation and staining were carried out, as described previously.

## Phosphoflow experiment

Enriched NK cells were stimulated with rmIL-15, rmIL-2, N-acetyl L-cysteine (NAC; Sigma-Aldrich), or hydrogen peroxide ($H_2O_2$; Sigma-Aldrich) in RPMI + 10% FBS for 5 min (BD Biosciences) at 37°C. Cells were then immediately fixed by adding an equal volume of two times diluted warm BD Cytofix Fixation Buffer (BD Biosciences) for 10 min at 37°C, permeabilized with 0.1% Triton X-100 for 5 min on ice, and barcoded with or without a Pacific Blue or Alexa Fluor 700 NHS ester dye (Invitrogen) at room temperature for 15 min. After three washes with RPMI + 10% FBS, differently barcoded samples were pooled and permeabilized with dropwise added Perm Buffer III (BD Biosciences) at 4°C for 30 min. Thereafter, samples were either kept at −20°C overnight or immediately stained with anti–phospho-antibodies, which was performed at room temperature for 1 h. Antibodies (BD Pharmingen) used to detect phosphorylated signalling molecules are pSTAT4 (pY693, 38/p-Stat4), pSTAT5 (pY694, 47/Stat5), pp38 (pT180/pY182, 36/p38), pERK (pT202/pY204, 20A), pAKT (pS473, N89-61), pmTOR (pS2448, O21-404), pLCK (pY505, 4/LCK), pSLP-76 (pY128, J141-668.36.58), pPLC-γ2 (pY759, K86-689.37), pZAP70 (pY319)/pSYK (pY352, 17A/P-ZAP70), pSTAT3 (pY705, 4/P-STAT3), pJNK (pT183/pY185, N9-66), pSRC (pY418, K98-37), p4EBP1 (pT36/pT45, M31-16), and pS6 (pS235/pS236, N7-548). In some experiments, after IL-15 stimulations, cells were washed three times with ice-cold PBS and incubated at 37°C for up to 3 h before being subjected to fixation,

Triton X-100 permeabilization, barcoding, Perm Buffer III permeabilization, and antibody staining.

## ROS staining

Freshly enriched NK cells were stimulated with IL-15, NAC, or $H_2O_2$ for indicated periods of time and then stained at a concentration of 1.5 × 10^6 cells per ml with 0.1 μM CM-$H_2$DCFDA (Invitrogen) in HBSS for 15 min at 37°C. The cells were then washed once with HBSS and incubated in HBSS at 37°C for another 15 min for esterases to act. Finally, surface staining was performed before the ROS level was measured by flow cytometry.

## Protein tyrosine phosphatase activity assay

Freshly enriched NK cells were lysed using 25 mM imidazole HCl (pH 7.0), 1 mg/ml BSA, 1% Triton X-100, 1 mM EDTA, 1 mM DTT, 1 mM phenyl-methylsulfonyl fluoride. Protein tyrosine phosphatase activity of murine NK cell lysate was measured using a kit from the Tyrosine Phosphatase Assay System (Promega). Briefly, endogenous phosphates were removed from cell lysate by passing through Sephadex columns twice. 250,000 NK cells were used per reaction, and each reaction was performed in duplicates. Phosphate-free lysate was incubated with indicated concentrations of $H_2O_2$ for 1 h at 37°C in the presence of phosphorylated peptides provided by the kit. The reactions were stopped using the molybdate dye, and the colour was developed at RT for 15 min. The amount of molybdate–phosphate complex was measured at 590 nm absorbance using the Infinite 200 Pro plate reader (Tecan).

## Seahorse metabolic flux analyser

The analysis of the OCR and extracellular acidification rate (ECAR) of enriched NK cells was performed following an established protocol (42). Briefly, 200,000–250,000 enriched NK cells were either directly seeded onto Seahorse plates and stimulated with IL-15 for 20 min or 22 h, followed by three washes before seeding. A Seahorse XFe-96 Analyzer (Seahorse Biosciences) was used for all the experiments. Seahorse plates were coated with poly-L-lysine (Sigma-Aldrich) the night before cell seeding. NK cells were resuspended in the Seahorse medium (RPMI supplemented with 25 mM glucose, 1 mM sodium pyruvate, 2 mM glutamine, and 1% FBS) and seeded onto the coated plate using spinning in a swinging bucket with a zero deceleration. ECAR and OCR were measured upon injections of oligomycin (2 μM), FCCP (1 μM), and rotenone (100 nM) (Sigma-Aldrich). Where indicated in Fig 4A, IL-15 to 100 ng/ml final concentration, or Seahorse medium without FBS (used for untreated controls), was injected onto the Seahorse plate.

## Killing assay

YAC-1 cells were labelled with 0.5 μM carboxyfluorescein succinimidyl ester (CFSE) (Invitrogen) for 20 min at room temperature. Subsequently, the labelled YAC-1 cells were incubated with enriched NK cells at effector-to-target ratios as indicated for a total time of 4 h 30 min. Anti-CD107a antibody (1D4B; BD Biosciences) was present from the beginning of the coculture. After the assay, the cells were stained with anti-NK1.1 (PK136; BioLegend), anti-CD3

(145-2C11), and LIVE/DEAD Fixable Aqua Dead Cell Stain Kit (Invitrogen). The percentages of live target cells were calculated based on the numbers of cells negative for the live/dead marker in each sample over total CFSE positive cells in the samples without NK cells. All experiments were run in triplicates.

## Statistical analysis

Data were analysed and plotted with GraphPad Prism version 6 for Mac OSX (GraphPad Software, www.graphpad.com).

## Supplementary Information

## Acknowledgements

We thank the PKL4 animal facility (Karolinska Institute, Huddinge) staff for taking care of the mice and helping out with blood samplings. We also thank Hongya Han, Heinrich Schlums, Yenan Bryceson, Arnika Kathleen Wagner, and Michael Chrobok for reagents, useful scientific inputs as well as technical helps in phosphoflow experiments. We would like to acknowledge Iyadh Douagi and the MedH Flow Cytometry core facility, supported by a joint core facility grant from the Karolinska Insitutet and the Stockholm City Council, for providing cell analysis services. We also thank the Centre for Haematology and Regenerative Medicine (HERM), Karolinska Institute, for providing a great facility and scientific environment. Funding: This work was funded by grants to P Höglund from the Swedish Research Council, the Swedish Cancer Society, Region Stockholm, Radiumhemmets Forskningsfonder, Aroseniusfonden, and Karolinska Institutet. TT Luu and S Ganesan were both supported by a Karolinska Institute Doctoral (KID) grants from Karolinska Institutet. L Schmied was supported from the Swiss National Science Foundation (SNSF): P400PM_183909.

## Author Contributions

TT Luu: conceptualization, data curation, formal analysis, validation, investigation, visualization, methodology, project administration, and writing—original draft, review, and editing.
L Schmied: data curation, formal analysis, investigation, methodology, and writing—original draft, review, and editing.
N-A Nguyen: data curation, formal analysis, investigation, and methodology.
C Wiel: data curation, formal analysis, investigation, methodology, and writing—review and editing.
S Meinke: conceptualization, resources, data curation, validation, and investigation.
D Mohammad: formal analysis, methodology, and writing—review and editing.
M Bergö: formal analysis, methodology, and writing—review and editing.
E Alici: funding acquisition, investigation, and writing—review and editing.
N Kadri: conceptualization, investigation, and writing—review and editing.
S Ganesan: formal analysis, methodology, and writing—review and editing.

P Höglund: conceptualization, formal analysis, supervision, funding acquisition, investigation, project administration, and writing—original draft, review, and editing.

## Conflict of Interest Statement

The authors declare that they have no conflict of interest.

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
