## [Reviewer comments · Life Science Alliance]

Life Science Alliance

Short-term IL-15 priming leaves a long-lasting signaling imprint in mouse NK cells

Thuy T. Luu, Laurent Schmied, Ngoc-Anh Nguyen, Clotilde Wiel, Stephan Meinke, DARA MOHAMMAD, Martin Bergö, Evren ALICI, nadir KADRI, Sridharan Ganesan, and Petter HÖGLUND
DOI: <https://doi.org/10.26508/lsa.202000723>

Corresponding author(s): Petter HÖGLUND, Karolinska Institutet

Review Timeline:

Submission Date:	2020-03-31
Editorial Decision:	2020-04-20
Revision Received:	2020-12-26
Editorial Decision:	2021-01-14
Revision Received:	2021-01-22
Accepted:	2021-01-25

Scientific Editor: Shachi Bhatt

Transaction Report:

April 20, 2020

Re: Life Science Alliance manuscript #LSA-2020-00723

Prof. Petter HÖGLUND
Karolinska Institutet
Hematology and Regenerative Medicine
Medicinaren 25/Neo
SE-141 86 Huddinge
Stockholm 14183
Sweden

Dear Dr. Höglund,

Thank you for submitting your manuscript entitled "Short-term IL-15 priming leaves a long-lasting imprint in NK cells independent of a metabolic switch" to Life Science Alliance. The manuscript was assessed by expert reviewers, whose comments are appended to this letter.

As you will see, the reviewers have a somewhat mixed view on your work, and think that the value provided to others is rather small at the moment. However, they also provide constructive input on how to strengthen your work and to increase its relevance. Should you be willing to revise your work along the lines suggested by the reviewers, we would be happy to consider your study further for publication here. Importantly, the kinetic analysis should get extended, the requested controls added, and the data representation must allow judging the robustness of the work. Further, at least an in vitro demonstration of the potential physiological significance of short-term NK cell activation is needed.

The typical timeframe for revisions is three months. We are aware that many laboratories cannot function fully during the current COVID-19/SARS-CoV-2 epidemic and therefore encourage you to take the time necessary to revise the manuscript to the extent requested above. We will extend our 'scoping protection policy' to the full revision period required. If you do see another paper with related content published elsewhere, nonetheless contact me immediately so that we can discuss the best way to proceed.

Please note that papers are generally considered through only one revision cycle, so strong support from the referees on the revised version is needed for acceptance.

Thank you for this interesting contribution to Life Science Alliance. We are looking forward to receiving your revised manuscript.

Sincerely,

B. MANUSCRIPT ORGANIZATION AND FORMATTING:

*****IMPORTANT:** It is Life Science Alliance policy that if requested, original data images must be made available. Failure to provide original images upon request will result in unavoidable delays in publication. Please ensure that you have access to all original microscopy and blot data images

before submitting your revision.***

Reviewer #1 (Comments to the Authors (Required)):

Review of manuscript No. LSA-2020-00723 by Luu et al.

The manuscript makes the interesting observation that short term exposure of mouse NK cells to IL-15 has a transient enhancing effect on the activity of NK cells. The manuscript is well written and makes an important contribution to our understanding of NK cell regulation. I applaud the authors for using barcoding to carefully compare different samples. However, I also have a few remarks:

Major points:

1. While the effects of short-term IL-15 exposure on IFN γ production, Ca flux, degranulation and cytotoxicity are convincing, the functional explanation is a bit less convincing. Due to the careful analysis employing barcoding of different samples the authors can show small effects of IL-15 exposure on the baseline-phosphorylation of several signaling molecules that are typically not associated with IL-15 signaling. However, while the authors infer from these findings that this is the reason why the stimulation of activating receptors is more efficient in IL-15 exposed NK cells, they don't address this experimentally. This should be done. For example: Is the phosphorylation of LCK, SLP-76 or other signaling molecules stronger upon NK1.1 stimulation when comparing IL-15 pre-exposed to non-exposed NK cells?
2. In order to stimulate NK cells with IL-15 for only a few minutes, the authors performed extensive washing in order to remove the IL-15 before performing their analysis. However, I am wondering if IL-15 can effectively be removed from the IL-15 receptor by washing. Some of the functional assays in figure 1 or figure 6 take several hours. If the IL-15 would remain bound to its receptor, the NK cells would continue to be stimulated during this time. One important control would be to employ the JAK3 inhibitor after 5 or 20 minutes of IL-15 exposure to disable IL-15 signaling at specific time points. This is especially relevant for the experiment shown in figure 6.
3. Figure 4: The authors speculate that ROS production could be one way how IL-15 enhances activating receptor signaling. While the authors are already careful in their interpretation of their data, I am puzzled by the clear inhibitory effect of NAC on STAT5 phosphorylation. To me this is an indication that NAC directly inhibits early IL-15 signaling. Therefore, it is no surprise that IL-15 mediated enhancement of other activating receptors can be blocked by NAC and this does not indicate that this enhancement is mediated by ROS. This should at least be discussed.

Minor points:

4. It should be mentioned in the title or the abstract that the studies were performed with mouse NK cells.
5. Discussion line 275: The authors discuss that signaling molecules such as PLC-g or ZAP-70 may directly associate with the IL-15 receptor. However, their data in figure S3 show that these two signaling molecules are not targeted by IL-15 signaling. This should be mentioned.
6. For several figures it is not mentioned in the graph or the legend how the cells were stimulated or for how long. Examples: Figure 1A, Figure 2, Figure S2A: How were the cells stimulated? (NK1.1?); Figure 2A, Figure 6, Figure S2B, C, Figure S4I, : How long were the cells pre-stimulated with IL-15?

Reviewer #2 (Comments to the Authors (Required)):

In this work, Luu et al describe the impact of short-term IL-15 exposure on murine NK cells. The duration of cytokine exposure is an understudied facet of the cellular immune response, so the subject of this study represents a desirable contribution to the field.

The authors first show that short-term IL-15 exposure enhances both the cytotoxicity and cytokine production of activated murine NK cells. They provide evidence that this enhancement is concomitant with a unique pattern of phosphorylation among signaling molecules in the activation pathway. Importantly, the authors provide evidence that short-term IL-15 exposure fails to trigger the metabolic re-programming associated with long-term exposure. Through blockade of JAK3, the authors show that the effects of short-term IL-15 exposure can be abrogated, providing a mechanistic basis for further study.

That such a brief exposure to IL-15 can influence NK cell activation signaling and downstream function, makes for a fascinating story. However, the work as presented is disappointing in scope, because most experiments do not directly compare short-term to long-term IL-15 exposure, instead merely comparing short-term exposure to no exposure. In addition, the effects of short-term exposure on activation signaling would have benefited by comparison to another cytokine or positive control stimulation. However, these shortcomings do not preclude publication.

The lack of metabolic reprogramming for short-term IL-15 exposure is well supported by a comparison to the effects of long-term exposure. The impact of IL-15 on NK cell functions is adequately supported, as are the activation signaling claims, with the caveat that the authors must address the use of their cellular barcoding procedure, and verify that barcoding does not influence the results of specific experiments, as detailed below. In the absence of controls for the barcoding procedure throughout, it is unclear whether the authors have met a minimum standard of reproducibility with their data. The writing and grammar also need some additional work.

MAJOR ISSUES:

1. In Figure 1 the authors combined cells from multiple mice into one sample, extracting individual data for each mouse based on cell trace markers. This is very unusual. If this were a necessary precaution to "reduce variation in experimental conditions," then it suggests that the protocols being used are not optimized and/or equipment is not being maintained properly. The authors should provide a stronger rationale for the barcoding used in Figure 1. If this technique were really necessary, then pooling data from multiple experiments would be inappropriate throughout the study, as the expected deviation between experiments would likely be greater than that expected between individual spleens harvested on the same day.

Because cell trace affects the surface of NK cells, this approach introduces another variable in the experiments and in the analyses. The authors should also provide a supplemental figure showing that various doses of cell trace do not affect the downstream assays where they are used. This can be accomplished with the existing data for Figure 1, by showing that the sample with the greatest response is not consistently also the sample that received the greatest or least amount of cell trace.

2. The authors also appear to "barcode" samples in a manner that corresponds to different doses of IL-15 (Fig. 6C). In this case, the barcoding coincides with the independent variable being studied. The authors must describe in detail the process of barcoding that occurred for experiments in Figure 6, and the rationale behind it. In addition, the authors should include a supplemental figure

demonstrating that the barcoding of untreated samples as in Figure 6, does not affect Flou-4 readings or pSTAT5 MFI in untreated samples. In addition, if the barcoding was performed prior to IL-15 treatment, the authors must show that there is no significant difference in the amount of cell death between the doses of barcoding dye or ester used. Otherwise, it is difficult to believe that the data in Figure 6 and perhaps elsewhere is due to IL-15 alone, and not some combined effect of IL-15 and the barcoding protocol.

3. The breed of mouse used is not indicated in M&M. The authors should clarify this point exactly. Also, please include the sex of the mice used throughout the study.

4. The figure legend for figure 1 states that NK cells were stimulated with either cytokine or cytokine/receptor complex, but no indication is given as to which was used in each experiment. The description for these experiments in the Results section only mentions IL-15. Please clarify. If data were pooled from cytokine and cytokine/complex experiments, then there is not enough data to support the claims in Fig. 1.

5. Throughout the manuscript, the type of statistical post-tests is not given. For instance, in Fig. 2 the Kruskal-Wallis test is not what is shown in Fig. 2B,D,F. The overall test may have been K-W, but the pairwise tests are Dunn's or an alternative test. The results of the overall ANOVA or K-W should be given in the figure legends, and the type of post-test performed should also be given throughout, in addition to the comparisons already shown. These factors are particularly important, as low numbers of mice and repeats are presented throughout.

6. The authors do not provide any evidence that their procedure for isolating NK cells is effective at isolating NK cells. A supplemental figure should be added to this effect, showing the % NK cells after isolation from the spleen.

7. In the Results on line 146 - "a small but consistent decrease of the respiratory capacity, as measured by maximal oxygen consumption rate (OCR) (Fig. S4A, B)." The figure does not show any significance for max OCR, so this interpretation should be changed or the significance should be shown.

MINOR ISSUES:

1. Line 378 - It is not clear why cells were washed with an ice-cold buffer then incubated at 37C. Why would cells be intentionally cold-shocked before an incubation, or warmed up before fixation? Please clarify.

2. Fig. 1: the concentration of IL-15 is not indicated in the results, nor in the figure legend, nor overall in the M&M for these experiments. If it is 100 ng/ml, please indicate this.

3. Figure 4 - please indicate the duration of IL-15 treatment on each panel of the figure, as the durations change from graph to graph.

GRAMMATICAL AND AESTHETIC ISSUES:

1. Fig. 4E-F, Fig. 5B - Circles are repeated for two groups in these graphs. Consider changing one of the groups to another shape, or all of the groups to the same shape. In addition, please consider

reducing the number of superfluous statistical comparisons shown, in order to highlight the key statistical comparison, that of IL-15 w/out NAC, vs IL-15 with 5,10,20mM NAC.

2. Throughout: the phrase "short-time" is used to describe the duration of IL-15 exposure. This should be changed to "short-term," as the phrase "short-time" is not really a common term. Examples include lines 116, 133, 185, 195, 206, etc. If, on the other hand, the authors are trying to make "short-time" a new descriptor for this procedure, then they should define it at the beginning of the Results section.

By Line number:

Line 58 - Should be "this effect" not "effects."

Line 71 - Consider removing "already" throughout the manuscript, as this is not a scientific term.

Line 99 - Should be "a 5-hour" or "5 hours of"

Figure 4 -How exactly were 22-hour stimulated NK cells compared to freshly isolated cells? Were cells taken from different mice, or were PBMC used? The number of mice used is not given for Figure 4 C,D.

Line 181 - Please add a sentence and provide a reference for the activity of N-acetyl cysteine.

Line 201 - This section is grammatically awkward. You might consider rearranging to something like: "the effect of IL-15 priming on IFN γ and degranulation was abolished at 200 ng/ml... "

Line 209-210 - This sentence doesn't make any sense.

Line 221 - This should be "as to how IL-15 primes..."

Line 310 - Consider using the term "housed" rather than "taken care of"

Line 311 - Consider simply "mice were six to ten weeks old."

Line 312 - Perhaps this should be "of the MHC class I molecules..."

Line 358 - Should this be "as indicated" ?

Line 409 - Perhaps "without brakes" is what was meant for the centrifugation, instead of "without breaks"?

Line 411 - This sentence is missing something, because it doesn't make sense as written. Please clarify.

Reviewer #3 (Comments to the Authors (Required)):

The work by Luu et al explore the phenomenon of NK cell priming by IL15. Based on the data presented the authors conclude that as little as 5 min to cytokine exposure is sufficient to give a long-lasting and reversible effect. Although the basic observation is interesting, and potentially of therapeutic importance, the work is very descriptive, and there is a critical lack of in vivo data or demonstration of relevance in humans.

1. Fig 1. The data shown should be extended to include a more detailed kinetics
2. The descriptive in vitro data should be complemented with data in vivo demonstrating physiological/pharmacological relevance of short-term NK cell activation
3. The current data are based on mouse data. It is essential that the authors show that similar phenomena are seen for human NK cells

Stockholm 2020-12-26

Dear Editor,

Please find below a point-by-point reply to the comments provided by the three reviewers on our manuscript LSA-2020-00723 by Luu et al. Our answers are set in red below following each reviewer's comments in black.

We are grateful to the editors and the reviewers for taking the time to study our data in such detail. The constructive critique and all suggestions have prompted us to perform several additional experiments, which has allowed us to reassess some conclusions and to modify the text accordingly.

The revised version of the paper contains 8 new figure items that we believe provide answers to most of the reviewer's comments. The novel display items are the following:

Fig. 3E,F
Fig. 6C,E
Fig. S1A,B
Fig. S2A,C,D
Fig. S3A,B
Fig. S4D;

We hope that our revision is sufficient to make our study suitable for publication in *Life Science Alliances*.

Sincerely yours,

Luu Thanh Thuy, first author
Petter Höglund, corresponding author

Point-by-point reply to the reviewer's comments

Reviewer #1:

The manuscript makes the interesting observation that short term exposure of mouse NK cells to IL-15 has a transient enhancing effect on the activity of NK cells. The manuscript is well written and makes an important contribution to our understanding of NK cell regulation. I applaud the authors for using barcoding to carefully compare different samples. However, I also have a few remarks:

Major points:

1. While the effects of short-term IL-15 exposure on IFN γ production, Ca flux, degranulation and cytotoxicity are convincing, the functional explanation is a bit less convincing. Due to the careful analysis employing barcoding of different samples the authors can show small effects of IL-15 exposure on the baseline-phosphorylation of several signaling molecules that are typically not associated with IL-15 signaling. However, while the authors infer from these findings that this is the reason why the stimulation of activating receptors is more efficient in IL-15 exposed NK cells, they don't address this experimentally. This should be done. For example: Is the phosphorylation of LCK, SLP-76 or other signaling molecules stronger upon NK1.1 stimulation when comparing IL-15 pre-exposed to non-exposed NK cells?

Reply

Thank you for this question, which is of great relevance and interest. To perform the experiment requested by the reviewer, we labeled NK cells with biotin-conjugated anti-NK1.1 or anti-NKp46 antibodies, stimulated them with IL-15 for 20 minutes, crosslinked the biotinylated antibodies using streptavidin and finally stained the cells using antibodies against phosphorylated signaling proteins. It was necessary to modify our protocol in the initial submission to make use of biotin/streptavidin instead of primary/secondary antibodies for crosslinking. The reason was to avoid unspecific binding of the anti-phosphoprotein antibodies to the crosslinked NK cells.

Biotin-conjugated anti-NK1.1 from two different companies (BioLegend and eBiosciences) unfortunately failed to activate NK cells using this protocol (data not shown). In contrast, using biotinylated antibody against NKp46 (another ITAM-dependent activating receptor on NK cells with similar signaling properties as NK1.1) was successful. NKp46 stimulation resulted in a reproducible increase in phosphorylation of the early signaling molecules ZAP70/SYK, LCK, SLP76 and PLC- γ after streptavidin crosslinking. Importantly, NK cells primed with IL-15 displayed an enhanced phosphorylation response after NKp46 crosslinking compared to NK cell that were not primed, which supports our conclusions and fulfils the prediction from the reviewer (Fig. 3E). A new text section has been added in the results part to describe these experiments.

Interestingly, this new setup also allowed us to note a weak, but statistically significant, difference between IL-15 alone and IL-15+NKp46 in the phosphorylation of STAT-5 (Fig. 3F). This result was unexpected given that STAT-5 is not known to be a part of the NKp46 signalling pathway, even though there was a small, yet significant, increase in pSTAT5 upon NKp46 stimulation (Fig. 3F). A similar result was obtained for pERK1/2 (Fig. 3F). This data further suggests that the cytokine signalling pathway and the ITAM pathway intersects in a crosstalk. We are grateful to the reviewer for suggesting this experiment, which allowed us to reach deeper into the mechanistic conclusions of our study.

2. In order to stimulate NK cells with IL-15 for only a few minutes, the authors performed extensive washing in order to remove the IL-15 before performing their analysis. However, I am wondering if IL-15 can effectively be removed from the IL-15 receptor by washing. Some of the functional assays in figure 1 or figure 6 take several hours. If the IL-15 would remain bound to its receptor, the NK cells

would continue to be stimulated during this time. One important control would be to employ the JAK3 inhibitor after 5 or 20 minutes of IL-15 exposure to disable IL-15 signaling at specific time points. This is especially relevant for the experiment shown in figure 6.

Reply

This is also a very interesting comment, but also quite complex to address. In fact, it includes two separate questions that need to be tested using different approaches. The first is if IL-15 remains bound to the NK cell surface after washing. This question might be tested using a sensitive FACS-based assay using reagents specific for the IL-15/IL-15R complex. Alternatively, a system based on photosensitive IL-15 or a proteomics approach might be developed. The second question is to which extent JAK3 is required to maintain the enhances NK cell activation in primed NK cells. This may or may not depend on remaining IL-15 at the cell surface and thus must be studied separately.

We have made a few initial attempts to stain for IL-15 at the NK cell surface after washing using FACS, but rapidly found that sensitivity was low and significant technical developments will be required to set up a robust test system. The other systems also require novel test platforms that we do not have access to at the moment. We are therefore unable to provide any novel data to prove or disprove that IL-15 remains at the NK cell surface.

After careful consideration, we also decided not to embark on a dissection of the role of JAK3 in the priming effect for this study. Testing this properly require that the inhibitor is applied at all steps of the assay, separately and in combination, for example before priming, during priming, after priming and during the functional assays. This major question will be studied in a follow-up, in which we also will attempt to study the proposed receptor crosstalk using biochemistry on primary cells and on cell lines (to secure enough protein to study which is difficult on primary murine NK cells).

We wish to point an aspect of our iJAK3 data not emphasized enough before, which is the lack of inhibitory effect of iJAK3 on NK1.1 stimulation of unprimed cells (Fig. 5C). These data emphasize that JAK3 is only brought into the picture when IL-15 is there to prime the cells, a conclusion that is now pointed out more clearly in the revised manuscript.

3. Figure 4: The authors speculate that ROS production could be one way how IL-15 enhances activating receptor signaling. While the authors are already careful in their interpretation of their data, I am puzzled by the clear inhibitory effect of NAC on STAT5 phosphorylation. To me this is an indication that NAC directly inhibits early IL-15 signaling. Therefore, it is no surprise that IL-15 mediated enhancement of other activating receptors can be blocked by NAC and this does not indicate that this enhancement is mediated by ROS. This should at least be discussed.

Reply

Thanks for pointing this out. It is correct that we cannot exclude that NAC might inhibit IL-15 signaling at a stage preceding the crosstalk rather than inhibiting the crosstalk as such. We have added one sentence in the results section regarding this interpretation.

Minor points:

4. It should be mentioned in the title or the abstract that the studies were performed with mouse NK cells.

Reply

Short-term IL-15 priming was seen also for human NK cells and because this was also a question from reviewer 3, we have now included human NK cell data in the revised paper (Fig. S2C and D).

However, since the human data only constitutes a minor part of our paper, we decided to follow the suggestion of this reviewer and added "mouse" in our title.

5. Discussion line 275 (line 300 in the revised manuscript): The authors discuss that signaling molecules such as PLC- γ or ZAP-70 may directly associate with the IL-15 receptor. However, their data in figure S3 show that these two signaling molecules are not targeted by IL-15 signaling. This should be mentioned.

Reply

The new data provided by the revision confirmed the data in Fig. S3 that the steady state phosphorylation of PLC- γ and ZAP70/SYK was not increased upon IL-15 stimulation alone (Fig. 3E). However, IL-15 stimulation nevertheless enhanced ITAM-dependent activation of these molecules, which lead us to conclude that IL-15 stimulation might cross-talk with activating signaling also via ZAP70/SYK and PLC- γ . A modifying sentence was included in the discussion.

6. For several figures it is not mentioned in the graph or the legend how the cells were stimulated of for how long. Examples:

Figure 1A, Figure 2, Figure S2A: How were the cells stimulated? (NK1.1?); Figure 2A, Figure 6, Figure S2B, C, Figure S4I : How long were the cells pre-stimulated with IL-15?

Reply

Thank you for pointing this out. The requested information has now been added to the legends.

Reviewer #2 (Comments to the Authors (Required)):

In this work, Luu et al describe the impact of short-term IL-15 exposure on murine NK cells. The duration of cytokine exposure is an understudied facet of the cellular immune response, so the subject of this study represents a desirable contribution to the field.

The authors first show that short-term IL-15 exposure enhances both the cytotoxicity and cytokine production of activated murine NK cells. They provide evidence that this enhancement is concomitant with a unique pattern of phosphorylation among signaling molecules in the activation pathway. Importantly, the authors provide evidence that short-term IL-15 exposure fails to trigger the metabolic re-programming associated with long-term exposure. Through blockade of JAK3, the authors show that the effects of short-term IL-15 exposure can be abrogated, providing a mechanistic basis for further study.

That such a brief exposure to IL-15 can influence NK cell activation signaling and downstream function, makes for a fascinating story. However, the work as presented is disappointing in scope, because most experiments do not directly compare short-term to long-term IL-15 exposure, instead merely comparing short-term exposure to no exposure. In addition, the effects of short-term exposure on activation signaling would have benefited by comparison to another cytokine or positive control stimulation. However, these shortcomings do not preclude publication.

The lack of metabolic reprogramming for short-term IL-15 exposure is well supported by a comparison to the effects of long-term exposure. The impact of IL-15 on NK cell functions is adequately supported, as are the activation signaling claims, with the caveat that the authors must address the use of their cellular barcoding procedure, and verify that barcoding does not influence the results of specific experiments, as detailed below. In the absence of controls for the barcoding procedure throughout, it is unclear whether the authors have met a minimum standard of reproducibility with their data. The writing and grammar also need some additional work.

MAJOR ISSUES:

1. In Figure 1 the authors combined cells from multiple mice into one sample, extracting individual data for each mouse based on cell trace markers. This is very unusual. If this were a necessary precaution to "reduce variation in experimental conditions," then it suggests that the protocols being used are not optimized and/or equipment is not being maintained properly. The authors should provide a stronger rationale for the barcoding used in Figure 1. If this technique were really necessary, then pooling data from multiple experiments would be inappropriate throughout the study, as the expected deviation between experiments would likely be greater than that expected between individual spleens harvested on the same day.

Reply

Quantifying phosphorylation of proximal signaling molecules after activation of immune cells is difficult for reasons of sensitivity. This was also the case in our model, in particular since we were using naïve ex vivo NK cells. We developed the barcoding protocol as a strategy to reduce inter-sample variation, arguing that minimizing this variation would be beneficial for showing small but consistent differences. We believe that this has been a successful approach. Similar results would most likely have been possible to generate in a more classical setup with samples analyzed in parallel, but would have required many more experiments to achieve significant differences.

With regard to the pooling, we still need to combine the data from several experiments to allow for statistical evaluations. Because each comparison is paired (being performed from an NK cell sample from one mouse), it was possible to normalize the control in each paired experiment and relate the value for the experimental sample to this normalized control. In this way, we could combine the experiments and generate reliable statistics. We hope that this explanation provides an acceptable account for why we used a barcoding strategy, and why it has been beneficial for our setup rather than the opposite.

Because cell trace affects the surface of NK cells, this approach introduces another variable in the experiments and in the analyses. The authors should also provide a supplemental figure showing that various doses of cell trace do not affect the downstream assays where they are used. This can be accomplished with the existing data for Figure 1, by showing that the sample with the greatest response is not consistently also the sample that received the greatest or least amount of cell trace.

Reply

This is a very valid comment. We have made sure that the dye does not affect the NK cell functions and have now included an example of our validation experiment in the revised version of the paper (Fig. S1A).

2. The authors also appear to "barcode" samples in a manner that corresponds to different doses of IL-15 (Fig. 6C). In this case, the barcoding coincides with the independent variable being studied. The authors must describe in detail the process of barcoding that occurred for experiments in Figure 6, and the rationale behind it. In addition, the authors should include a supplemental figure demonstrating that the barcoding of untreated samples as in Figure 6, does not affect Flou-4 readings or pSTAT5 MFI in untreated samples.

Reply

We appreciate these technical concerns and have included data in the revised version to control for these concerns (Fig. S2A; S3B).

In addition, if the barcoding was performed prior to IL-15 treatment, the authors must show that there is no significant difference in the amount of cell death between the doses of barcoding dye or ester used. Otherwise, it is difficult to believe that the data in Figure 6 and perhaps elsewhere is due

to IL-15 alone, and not some combined effect of IL-15 and the barcoding protocol.

Reply

For phospho-flow experiments, the barcoding was performed after IL-15 stimulation. For the IFN- γ and Calcium flux experiments, the barcoding was done before the stimulation. The impact of the barcoding on IFN- γ and CD107 responses and the cell death is shown in Fig. S1B.

3. The breed of mouse used is not indicated in M&M. The authors should clarify this point exactly. Also, please include the sex of the mice used throughout the study.

Reply

The mice used are on an inbred C57Bl/6 background, deficient for the classical MHC class I genes H2K^b and H2D^b but transgenic for the H2D^d gene. These mice were previously generated in our laboratory to provide a well-characterized genetic background for studies of NK cell education (see e.g. Johansson et al., *J. Exp. Med.* 201, 1145, 2015; Johansson et al., *PlosOne* 4, e6046, 2010; Brodin et al., *Blood* 113, 2434, 2009; Brodin et al., *J. Immunol.*, 188, 2218, 2012). These "single-H2D^d" mice have fully functional NK cells that are all educated on the H2D^d gene. In each experiment, mice were sex and age-matched. This information is now included in the M&M.

4. The figure legend for figure 1 states that NK cells were stimulated with either cytokine or cytokine/receptor complex, but no indication is given as to which was used in each experiment. The description for these experiments in the Results section only mentions IL-15. Please clarify. If data were pooled from cytokine and cytokine/complex experiments, then there is not enough data to support the claims in Fig. 1.

Reply

This was our mistake, it should have been only "cytokine". The legend has now been edited to clarify this.

5. Throughout the manuscript, the type of statistical post-tests is not given. For instance, in Fig. 2 the Kruskal-Wallis test is not what is shown in Fig. 2B,D,F. The overall test may have been K-W, but the pairwise tests are Dunn's or an alternative test. The results of the overall ANOVA or K-W should be given in the figure legends, and the type of post-test performed should also be given throughout, in addition to the comparisons already shown. These factors are particularly important, as low numbers of mice and repeats are presented throughout.

Reply

Thank you for this important comment. The types of post-tests performed have now been included in the legend.

6. The authors do not provide any evidence that their procedure for isolating NK cells is effective at isolating NK cells. A supplemental figure should be added to this effect, showing the % NK cells after isolation from the spleen.

Reply

A panel is now included in Fig. S1A to indicate the purity of isolated NK cells.

7. In the Results on line 146 - "a small but consistent decrease of the respiratory capacity, as measured by maximal oxygen consumption rate (OCR) (Fig. S4A, B)." The figure does not show any significance for max OCR, so this interpretation should be changed or the significance should be shown.

Reply

The description (line 165 in the revised ms) is changed to “This experiment revealed a decreased level of proton leakage and a trend in decreased respiratory capacity, as measured by maximal oxygen consumption rate (OCR) (Fig. S4A, B).”

MINOR ISSUES:

1. Line 378 - It is not clear why cells were washed with an ice-cold buffer then incubated at 37C. Why would cells be intentionally cold-shocked before an incubation, or warmed up before fixation? Please clarify.

Reply

Line 403 in the revised manuscript. When setting up the assay, we found that the protocol that gave the best ROS staining was labelling with CM-H₂DCFDA at 37°C, washing with ice-cold HBSS buffer followed by a subsequent incubation for 15 minutes at 37°C. The ice-cold buffer was used to stop the labelling process as fast as possible.

Throughout the paper, we used ice-cold buffer to stop the cytokine stimulation period and wash away maximally unbound cytokines. The reason was to make the stimulation time more precisely controlled, which was important with short-time stimulations in a window from one minute to twenty minutes. Primed NK cells were then incubated at 37°C again in the downstream functional assays, such as NK1.1 stimulation followed by IFN- γ secretion, calcium flux or phosphorylation of signaling molecules.

For the phospho-flow experiments, cells were stimulated with IL-15 in RPMI plus 10 % FBS at 37C in a water bath, and after five or twenty minutes of stimulation fixed with warm fixation buffer for 10 minutes in the 37°C water bath. This is the standard phospho-flow protocol that we are using in our lab and it reproducibly detects phosphorylated signaling molecules.

2. Fig. 1: the concentration of IL-15 is not indicated in the results, nor in the figure legend, nor overall in the M&M for these experiments. If it is 100 ng/ml, please indicate this.

Reply

The concentration of IL-15 used in this figure was 100 ng/ml. Thanks for noticing that this information was missing. It is now included in the legend.

3. Figure 4 - please indicate the duration of IL-15 treatment on each panel of the figure, as the durations change from graph to graph.

Reply

The durations of IL-15 are now indicated in Fig. 4.

GRAMMATICAL AND AESTHETIC ISSUES:

1. Fig. 4E-F, Fig. 5B - Circles are repeated for two groups in these graphs. Consider changing one of the groups to another shape, or all of the groups to the same shape. In addition, please consider reducing the number of superfluous statistical comparisons shown, in order to highlight the key statistical comparison, that of IL-15 w/out NAC, vs IL-15 with 5,10,20mM NAC.

Reply

Thanks for this comment. We have followed your suggestion and changed the shapes as well as removed unnecessary statistical brackets.

2. Throughout: the phrase "short-time" is used to describe the duration of IL-15 exposure. This should be changed to "short-term," as the phrase "short-time" is not really a common term. Examples include lines 116, 133, 185, 195, 206, etc. If, on the other hand, the authors are trying to make "short-time" a new descriptor for this procedure, then they should define it at the beginning of the Results section.

Reply

We appreciate this note. We have changed all these notions to "short-term".

By Line number:

Line 58 - Should be "this effect" not "effects."

Line 71 - Consider removing "already" throughout the manuscript, as this is not a scientific term.

Line 99 - Should be "a 5-hour" or "5 hours of"

Reply

We have made changes on the corresponding lines in the new version of the manuscript.

Figure 4 -How exactly were 22-hour stimulated NK cells compared to freshly isolated cells? Were cells taken from different mice, or were PBMC used? The number of mice used is not given for Figure 4C,D.

Reply

Unstimulated NK cells were analyzed fresh on day one and the data stored. The cells were then put in culture with IL-15 and were analyzed again after 22 hours of incubation. In this way, data from NK cells analyzed fresh and after 22 hours from the same animal could be compared. The number of mice analyzed is now indicated in the legend.

Line 181 - Please add a sentence and provide a reference for the activity of N-acetyl cysteine.

Reply

Done.

Line 201 - This section is grammatically awkward. You might consider rearranging to something like: "the effect of IL-15 priming on IFN γ and degranulation was abolished at 200 ng/ml..."

Reply

Done (lines 234-235 in the revised manuscript)

Line 209-210 - This sentence doesn't make any sense.

Reply

Line 227-228 in the revised manuscript. We agree, there is something wrong with it. We have rewritten this sentence to better reflect the intended meaning.

Line 221 - This should be "as to how IL-15 primes..."

Line 310 - Consider using the term "housed" rather than "taken care of"

Line 311 - Consider simply "mice were six to ten weeks old."

Line 312 - Perhaps this should be "of the MHC class I molecules..."

Line 358 - Should this be "as indicated" ?

Reply

These five changes have been made on the corresponding lines in the new version of the manuscript.

Line 409 - Perhaps "without brakes" is what was meant for the centrifugation, instead of "without breaks"?

Reply

Line 433 in the revised manuscript. Many thanks for spotting this. We used the wrong word. The sentence has been changed to "using spinning in a swinging bucket with a zero deceleration".

Line 411 - This sentence is missing something, because it doesn't make sense as written. Please clarify.

Reply

Line 435 in the revised manuscript. The sentence has been changed to better reflect the intended meaning.

Reviewer #3 (Comments to the Authors (Required)):

The work by Luu et al explore the phenomenon of NK cell priming by IL15. Based on the data presented the authors conclude that as little as 5 min to cytokine exposure is sufficient to give a long-lasting and reversible effect. Although the basic observation is interesting, and potentially of therapeutic importance, the work is very descriptive, and there is a critical lack of in vivo data or demonstration of relevance in humans.

1. Fig 1. The data shown should be extended to include a more detailed kinetics

Reply

We understand the wish to extend the analysis of functional responses after additional time points of IL-15 priming. However, we do not think that including such additional data is necessary for our paper to provide novel and important knowledge. Priming following long-term stimulations have been demonstrated in numerous studies. Therefore, we deliberately choose at the start of this project to explore the mechanisms of short-term priming as such and not to make a detailed comparison with longer priming periods. Including additional time points at this stage would have taken the study too far both in scope and time and we have therefore decided not to expand the paper in this direction. An extended comparison of the mechanisms that differentiate short-term and long-term priming (some of which are indicated in the Seahorse experiments in this study) will be considered as a follow-up.

2. The descriptive in vitro data should be complemented with data in vivo demonstrating physiological/pharmacological relevance of short-term NK cell activation

In vivo experiments to support the roles of short-term IL-15 stimulation is very challenging and we have not yet come up with a feasible and robust model system to proceed in this direction. To provide more information in the *in vitro* model with the closest physiological relevance, YAC-1 killing, we repeated this assay using NK cells that were stimulated with IL-15 for 5 minutes in addition to the 20 minutes time point that was reported in the initial submission. Consistent with the data in Fig. 1E in the paper, IL-15 priming for 20-minutes increased killing capacity of NK cells towards YAC1 cells (Figure 1 in the rebuttal letter). When 5 minutes of priming was tested, we found that 3/5 mice displayed an increase in cytotoxicity while 2/5 mice did not (Figure 1 below). We take this result to indicate that 5 minutes of priming of a complex cytotoxicity response is at the border time-wise regarding kinetics. Following up on how various effector cell responses are induced and with which

kinetics will be of great interest to dissect this in a future study.

[Figure removed by editorial staff per authors' request].

We also decided to extend our data in Figure 6 of the paper regarding the long-lasting effect of IL-15 priming to complement the calcium flux and STAT-5 phosphorylation data with an additional more relevant effector function. In the revised Figure 6, we have now also included data on IFN- γ secretion, showing that this effector cell response persists after 20 minutes of IL-15 priming, washing, 3 hours of rest and assay time for cytokine secretion. Even if this experiment does not satisfy the demand for in vivo evidence, we believe it adds to the message of physiological relevance of short-term NK cell priming by IL-15.

3. The current data are based on mouse data. It is essential that the authors show that similar phenomena are seen for human NK cells

We thank the reviewer for this comment. We have found that also human cells can be primed with short-term IL-15 stimulation. We have included such data in the revised version, showing that one minute of being in contact with IL-15 augmented the basal intracellular calcium level (Fig. S2C) and the calcium flux of human NK cells in response to CD16 stimulation (Fig. S2D).

January 14, 2021

RE: Life Science Alliance Manuscript #LSA-2020-00723R

Prof. Petter HÖGLUND
Karolinska Institutet
Hematology and Regenerative Medicine
Medicinaren 25/Neo
SE-141 86 Huddinge
Stockholm 14183
Sweden

Dear Dr. HÖGLUND,

Thank you for submitting your revised manuscript entitled "Short-term IL-15 priming leaves a long-lasting signaling imprint in mouse NK cells". We would be happy to publish your paper in Life Science Alliance pending final revisions necessary to meet our formatting guidelines.

Along with the points listed below, please also attend to the following,

- please consult our manuscript preparation guidelines <https://www.life-science-alliance.org/manuscript-prep> and make sure your manuscript sections are in the correct order
- please make sure the author order in your manuscript and our system match and that there is no name discrepancy of the Authors between the system and manuscript file
- please upload both your main and supplementary figures as single files
- please add callout for Figure 6D to your main manuscript text

A. FINAL FILES:

B. MANUSCRIPT ORGANIZATION AND FORMATTING:

Sincerely,

Shachi Bhatt, Ph.D.
Executive Editor
Life Science Alliance
<https://www.lsjournal.org/>
Tweet @SciBhatt @LSAJournal

Reviewer #1 (Comments to the Authors (Required)):

The authors have sufficiently addressed all my concerns in the revised manuscript and have provided new data which significantly strengthen the manuscript. I have no more remarks.

Reviewer #3 (Comments to the Authors (Required)):

The authors have addressed most of the points that I raised, including addition of new data. Thus I find the work significantly improved. I am now convinced that the conclusions are justified based on the data presented.

January 25, 2021

RE: Life Science Alliance Manuscript #LSA-2020-00723RR

Prof. Petter HÖGLUND
Karolinska Institutet
Hematology and Regenerative Medicine
Medicinaren 25/Neo
SE-141 86 Huddinge
Stockholm 14183
Sweden

Dear Dr. HÖGLUND,

Thank you for submitting your Research Article entitled "Short-term IL-15 priming leaves a long-lasting signaling imprint in mouse NK cells". It is a pleasure to let you know that your manuscript is now accepted for publication in Life Science Alliance. Congratulations on this interesting work.

DISTRIBUTION OF MATERIALS:

Again, congratulations on a very nice paper. I hope you found the review process to be constructive and are pleased with how the manuscript was handled editorially. We look forward to future exciting submissions from your lab.

Sincerely,

Shachi Bhatt, Ph.D.

Executive Editor

Life Science Alliance

<https://www.lsjournal.org/>
